# Left Ventricular Unloading in Acute on Chronic Heart Failure: From Statements to Clinical Practice

**DOI:** 10.3390/jpm12091463

**Published:** 2022-09-06

**Authors:** Alice Sacco, Nuccia Morici, Jacopo Andrea Oreglia, Guido Tavazzi, Luca Villanova, Claudia Colombo, Laura Garatti, Michele Giovanni Mondino, Stefano Nava, Federico Pappalardo

**Affiliations:** 1”De Gasperis” Cardio Center, ASST Grande Ospedale Metropolitano Niguarda, 2011 Milan, Italy; 2IRCCS Fondazione Don Gnocchi, Dipartimento Cardio-Respiratorio, 2011 Milan, Italy; 3Department of Clinical-Surgical, Diagnostic and Paediatric Sciences, Unit of Anaesthesia and Intensive Care, University of Pavia Italy, 27100 Pavia, Italy; 4Anesthesia and Intensive Care, Fondazione Policlinico San Matteo Hospital IRCCS, Anestesia e Rianimazione I, 27100 Pavia, Italy; 5Cardiothoracic and Vascular Anesthesia and Intensive Care, AO SS. Antonio e Biagio e Cesare Arrigo, 15100 Alessandria, Italy

**Keywords:** cardiogenic shock, acute on chronic heart failure, mechanical circulatory support, escalation

## Abstract

Cardiogenic shock remains a deadly complication of acute on chronic decompensated heart failure (ADHF-CS). Despite its increasing prevalence, it is incompletely understood and therefore often misdiagnosed in the early phase. Precise diagnosis of the underlying cause of CS is fundamental for undertaking the correct therapeutic strategy. Temporary mechanical circulatory support (tMCS) is the mainstay of management: identifying and selecting optimal patients through understanding of the hemodynamics and a prompt profiling and timing, is key for success. A recent statement from the American Heart Association provided pragmatic suggestions on tMCS device selection, escalation, and weaning strategies. However, several areas of uncertainty still remain in clinical practice. Accordingly, we present an overview of the main pitfalls that can occur during patients’ management with tMCS through a clinical case. This case illustrates the strict interdependency between left ventricular unloading and right ventricular dysfunction in the case of low filling pressures. Moreover, it further illustrates the pivotal role of stepwise escalation of therapy in a patient with an ADHF-CS and its peculiarities as compared to other forms of acute heart failure.

## 1. Introduction

Cardiogenic shock (CS) is a heterogeneous clinical syndrome with increasing incidence and high mortality [1]. The two main etiologies of CS are acute myocardial infarction (AMI-CS) and acutely decompensated heart failure (ADHF-CS) [2]. The potential presentation of low cardiac output syndromes includes isolated hypotension with preserved organ perfusion; isolated organ hypoperfusion despite preserved blood pressure; and the combination of hypotension and hypoperfusion, which characterizes classical CS [3].

Recent consensus documents have highlighted that CS is defined by the presence of hypoperfusion although not necessarily requiring the presence of hypotension [1,4,5,6]. Three distinct clinical phenotypes [7] (I “non-congested,”, II “cardiorenal,” and III “cardiometabolic”) with specific characteristics were identified: they are associated with different risks of intra-hospital mortality, with phenotype III having the worst prognosis. Accordingly, a prompt identification of CS phenotypes could improve risk stratification and warrant adequate therapy. Two main differences characterize ADHF-CS as compared with AMI-CS: (i) functional reserve is persistently low and (ii) the loss of contractile reserve is irreversible. This latter feature implies that an “exit strategy” should be promptly defined in this setting and a timely escalation to the appropriate mechanical circulatory support (MCS) is mandatory in the case of clinical deterioration.

A recent scientific statement from the American Heart Association provided pragmatic suggestions on temporary mechanical circulatory support (tMCS) device selection, escalation, and weaning strategies, suggesting invasive monitoring to drive the management [8]. However, several areas of uncertainty still do exist and the early routine use of pulmonary catheter is limited in the current clinical practice.

Accordingly, we decided to review these key points through a clinical case of a patient with acute on chronic heart failure, complicated by CS.

This case describes, throughout a complete hemodynamic assessment, that right ventricular (RV) failure with pulmonary hypertension (PH) can occur because of ineffective unloading and would, paradoxically, dictate the need for LVAD implantation. The case description provides useful insights for clinicians managing ADHF-CS patients in an intensive cardiac care unit (ICCU).

## 2. CASE Description

A 62-year-old man was admitted to the Intermediate Cardiac Care with a diagnosis of post ischemic ADHF (Table 1). Ejection fraction (EF) was 20%. Left ventricle was severely dilated (end-diastolic diameter was 74 mm, 35 mm/cm^2^). Mitral regurgitation (MR) and tricuspid regurgitation (TR) were respectively moderate and mild; tricuspid annular plane systolic excursion (TAPSE) was 16 mm, RV fractional area change (RVFAC) was 29%, and RV tissue Doppler S velocity of the tricuspid annulus was 8 cm/s. He was congested and normotensive with chronic Stage 3b kidney disease, (Creat 1.75 mg/dL and urea 60 mg/dL, eGFR 40 mL/min).

The patient was started on furosemide, nitroprusside, and dopamine infusion. The patient deteriorated 5 days later (Table 1) and was transferred to ICCU: mean arterial pressure (MAP) was lower than 65 mmHg, lactates were 2.5 mmol, right atrial pressure (RAP) was 16 mmHg, and an intra-aortic balloon counterpulsation (IABP) was implanted. Over the next 6 days, hemodynamics significantly improved with MAP constantly over 65 mmHg, lactates below 2 mmol/L, and RAP 4–6 mmHg. MR and TR were mild, TAPSE increased to 18 mm, RVFAC to 40%, and RV tissue Doppler S velocity of the tricuspid annulus to 11 cm/s. Right heart catheterization (RHC) confirmed that pulmonary pressures and resistance were within normal limits and pulmonary arterial wedge pressure (PAWP) was 15 mmHg. The patient was therefore weaned from the IABP, but the clinical picture deteriorated over the next 24 h into a SCAI C, cardiometabolic phenotype III (Table 1) [7], and the IABP was re-inserted. The heart team decided to plan for placement of a durable left ventricular assist device (LVAD) implantation as a bridge to transplant. The patient’s hemodynamics further deteriorated with MAP of 50 mmHg, lactates 4 mmol/L, and RAP 14 mmHg. The echocardiographic picture did not show any significant difference compared to previous days, in particular, RVFAC was 34%. Rapid escalation with a percutaneous MCS device (Impella-CP; Abiomed) was deemed necessary (Table 1).

Hemodynamic stabilized with the device deployed at a support level of P8 (cardiac output-CO 3.4 L/min). Invasive monitoring showed CO 4.1 L/min, pulmonary pressures of 34/11/22 mmHg, PAWP of 9 mmHg, RAP of 0 mmHg, and indexed pulmonary vascular resistances (iPVR) of 1 Woods Units. Echocardiogram showed correct positioning of the Impella pump with a moderate MR, increased compared with previous days, and normal RV function.

Soon after the hemodynamic assessment post=Impella implantation, a hypotension occurred, which promptly responded to fluid challenge. The patient was awake and there were no signs of respiratory failure.

Ninety minutes after Impella activation, the patient developed hemodynamic collapse (Figure 1, panel A). The echocardiogram showed rightward shift of the interventricular septum (IVS) with normal RV dimensions and moderate MR. PAC revealed severe pulmonary hypertension; PAWP could not be measured [9]; RAP was 4 mmHg. An increase of RV stroke work index (RVSWI) was observed: from basal 3.8 g/m, (mPAP 18 mmHg and SVi 14.79 mL/m^2^/beat) to 8.5 g/m (mPAP 53 mmHg and SVi 13.11 mL/m^2^/beat). Moreover, there was a decrease of CO (from 3.2 to 2.9 L/min) despite an unchanged LVSWI [16 g/m (MAP 75 mmHg and SVi 15 mL/m^2^/beat) to 15.5 g/m (MAP 83 mmHg and SVi 13 mL/m^2^/beat)].

Sodium nitroprusside was discontinued due to hypotension. Intravenous fluids and red blood cell transfusions were delivered, and epinephrine was started at 0.12 mcg/kg/min.

After fluid challenge with 1000 cc of sodium chloride and 3 units of packed red blood cells, echocardiography confirmed right-sided shift of IVS, but increased dimensions of RV with reduced contractility.

The patient was intubated and inhalation of nitric oxide (20 ppm) was started, with a rapid normalization of pulmonary pressures and hemodynamic stabilization (Figure 1, panel B and C) allowing epinephrine weaning. The patient’s hemodynamics improved significantly over the next 6 h: complete lactates clearance, urine output was normal (2.2 cc/kg/hr), and SvO2 increased from 30 to 60%. Afterwards, the clinical picture further deteriorated (Figure 1, panel D and E) refractory to inotropes titration.

The Heart team discussed the possibility of VA-ECMO, but circulatory support upgrade was withheld because the chance of bridging to LVAD or orthotopic heart transplantation (OHT) appeared unlikely.

The patient expired 24 h after Impella implantation.

## 3. Discussion

The present case illustrates the challenges in promptly identifying and managing high-risk ADHF-CS phenotypes from drugs to escalating tMCS.

On admission, clinical evaluation was misled by normotension and as a consequence we overlooked the metabolic damage, missing the identification of CS phenotype III-cardiometabolic [7] and SCAI stage B [4]. Indeed, these high-risk patients may benefit from early advanced forms of mechanical circulatory support quickly bridged to LVAD without any evaluation for urgent heart transplantation, if not already listed. Moreover, PAC was delayed beyond 24 h from admission, and this may be associated with higher rates of death [10,11,12,13].

In this regard, the IABP was able to stabilize the patient and might have effectively bridged the patient to LVAD. However, the lack of an immediate long-term strategy produced a significant delay in the process of care. Furthermore, if temporary mechanical support escalation from IABP is required, we envision the role of an axillary Impella 5.0/5.5 as a potentially superior escalation strategy to fully unload the left ventricle and provide a higher level of systemic perfusion while assessing the ability of the right ventricle to tolerate high-forward flow from the LV, as a tolerance test for a durable LVAD [14,15].

In our patient, the Impella CP device was chosen over Impella 5.0/5.5 because we thought it could provide full cardiac support and due to its ease of insertion, did not need surgical involvement [16]. Additionally, we deemed venoarterial extracorporeal membrane oxygenation (VA-ECMO) not necessary, because our patient did not develop respiratory failure and hypoxia (Figure 2), although RV failure occurred at the latest stage.

Impella CP was able to adequately unload, but could not provide sufficient support for biventricular failure. The patient likely needed a biventricular support strategy early on, but this was not easily identified due to the RA pressure of zero. In fact, the residual LV function was overestimated due to the low filling pressures obtained with diuretics.

The following key points can be drawn from this clinical case.


**QUESTION 1: What is the role of filling pressures in patients with ADHF-CS?**


Patients with ADHF usually present with congestion and clinicians are usually prone to quickly start diuretics iv in order to reduce filling pressures and improve splanchnic congestion [5,17]. However, dilated ventricles are strictly dependent on preload and hypovolemia can precipitate cardiogenic shock. Once hemodynamics is hypotensive, the typical reaction is mechanical support as echocardiography shows severe LV dysfunction and mitral regurgitation.

The pharmacological strategies with diuretics and nitroprusside may have sub-optimal effect in the case of tight biventricular pre/afterload balance and delay the MCS implantation, leading to advanced stages of end-organ failure. In other terms, manipulating the hemodynamics with diuretics and nitroprusside might further delay mechanical support.

The ‘cold and dry’ ADHF-CS patients is an extremely challenging cohort since they play off the Starling curve with profoundly low cardiac output that is likely to worsen after volume resuscitation.

Additionally, the assessment of the right heart hemodynamic is particularly challenging since the RA pressure and RV function remains normal but highly susceptible to volume loading, which may precipitate RV failure.


**QUESTION 2: Which is the optimal mechanical circulatory support in ADHF?**


Among MCS, intra-aortic balloon counterpulsation (IABP) provides a modest hemodynamic support, potentially acting on ventricuo-arterial coupling (VAC) by reducing pre-systolic (end–diastolic) aortic pressure, left ventricular wall tension, and the rate of left ventricular pressure rise (dp/dt), and favouring coronary perfusion [18]. At the same time, IABP is easy to insert and manage and carries a low risk of vascular access complications. However, IABP prognostic impact as recently questioned by a randomized control study and meta-analyses [19,20] in acute myocardial infarction complicated by shock, downgrading its indication in the guidelines [17].

In contrast, there are limited data exploring the efficacy of IABP in patients with CS due to ADHF.

Recent studies and one review [21,22,23] showed that prolonged IABP support could lead to some degree of improvement in hemodynamics in patients with ADHF, being a valid bridge to LVAD or to HTx. An early detection of hypoperfusion and consequent timely IABP implantation might improve VAC by means of lowering arterial elastance [24]. Finally, a recent statement from American Heart Association supports the use of either IABP or Impella CP as initial left-sided tMCS device support in the presence of hypotension, elevated serum lactate, or other signs and symptoms of hypoperfusion, with evidence of LV failure [8].

Persistence of elevated LV filling pressures, pulmonary congestion, metabolic deterioration, and end-organ damage during IABP support should be criteria to promptly escalate the support.

We probably overlooked the severely dilated left ventricle (diastolic diameter was 74 mm, 35 mm/cm^2^). This aspect was certainly a drawback that should always be considered in the diagnostic/therapeutic pathway for the escalation or choice of Impella device, in particular if diameter is greater than 60 mm. In this case, it could be more appropriate to go directly to Impella 5.5 and, if not available, to VA-ECMO with CP as an unloading strategy (Figure 2).

After Impella CP was inserted, there was a minimal change in CO, but epinephrine levels were increased then the patient received multiple transfusions and volume loading, all of which severely decompensated the patient with biventricular failure and a dilated cardiomyopathy. This upgrade did not sufficiently increase the cardiac output to offload the LV, eliciting a dramatic change in the interplay between the ventricles. Moreover, the positive pressure ventilation with consequent reduction of venous return and increased afterload may have further altered the right side hemodynamic [25].

VA-ECMO is still considered the standard of MCS in patients with profound CS and hypoxia (INTERMACS 1 patient), but recently para-corporeal LVAD/biventricular assist device (BiVAD) have been taken into consideration more often.

ECMO has the advantage of being readily implanted in most patients and different clinical scenarios. However, high-dose inotropic support is often necessary, alongside IABP and/or LV venting, with the result of increasing the risk of local or systemic complications [26].

Para-corporeal VADs could have a role in selected adult patients, in particular in the case of biventricular failure [27]. The absence of the oxygenator can potentially lower the impact of inflammation and coagulation imbalance and offer a more physiologic blood flow both in uni- or bi-ventricular configurations. Moreover, the recovery can be successful enough to permit an improved patient’s fitness and nutritional status before OHT or LVAD.

We can assert that there were multiple levels of failure that may have been precipitated by the challenging assessment of biventricular function and pathophysiology, finally leading to an irreversible CS vicious cycle.

After the deterioration following IABP weaning, we should have immediately upgraded to a further hemodynamic supportas Impella and chosen a most powerful pump, Impella 5.0/5.5, with an upper body approach (Figure 2). This could have allowed resolution of organ damage and probably early extubation and mobilization [28].


**QUESTION 3: How much unloading is necessary to protect the right ventricle?**


Increased LV filling pressures in the setting of chronic LV failure increase RV afterload, which promotes RV remodeling. A recent study [29] described higher RV pressures and pulmonary artery pulsatility index in ADHF-CS compared to AMI-CS and further showed that loss of RV function is more strongly associated with death among ADHF-CS than AMI-CS patients. LV filling pressure is a major determinant of RV afterload and this highlights that LV failure is a common cause of RV failure [30,31]. As chronic heart failure is a long-standing state of low cardiac output, the amount of forward flow needed by an assist device could not necessarily be very high. As congestion is prominent, the right ventricle may benefit from the prompt reduction of LV filling pressures, even elicited by IABP, as a first step approach.

In this context, the thoughtful monitoring of RV hemodynamic with pulmonary catheter and function with, at least daily, echocardiography during mechanical unloading of the LV, is paramount. Among clinical predictors of RV failure, PAPI could be of clinical utility, but it may be challenging to interpret if the RAP is less than 10 or the PASP is >50 and with extremely severe RV/biventricular failure (stage E SCAI), as in our case.


**QUESTION 4: Should patients on temporary MCS receive inotropes?**


The use of adrenergic drugs is supported by a low level of evidence [16,32,33]. There are few studies in patients with cardiogenic shock and no studies specifically in the setting of biventricular failure and MCS. Considering the potential detrimental effect, especially exerted at high dosages [33,34], the general recommendation is to use them at the lowest dose and for the shorter time as possible [34].

Therefore, weaning from adrenergic agents is strongly suggested soon after the MCS is in place and failure to achieve it should prompt re-evaluation for ineffective unloading. The usual setting where the inotropes are maintained is in the absence of an ejecting pulse pressure (aortic valve opening) when a patient is on a continuous flow pump. However, this should be interpreted as ineffective offloading in the presence of increased afterload and, therefore, treated with flow titration and eventually MCS step-up approach.

Inotropes cut-off values are extremely valuable for clinical purposes, for example an inotropic score > 20 should warrant evaluation for device escalation [35].

## 4. Conclusions

According to recent data [8], the ADHF-CS in the presence of severe dilatation of LV, requires a rapid step-up device strategy.

RV modulates LV structure and function through diastolic and systolic interdependence in conditions of pressure and/or volume overload, and may be in turn affected by the induced changes in the left ventricle [36].

Moreover, RV contractility response to increased PAP is closely associated to its preserved systolic function/size: There is some ability of RV to increase its contractility with increases in afterload through homeometric autoregulation [37]. Our patient had a dysfunctional RV and presumably an unfavorable RV coupling, so that the acute and severe rise in the afterload elicited a further deterioration of RV function. Finally, circulatory support augmented RV preload, which was ineffectively handled by the declining RV performance.

When choosing a tMCS device, clinicians should take into account the etiology of shock (acute versus acute on chronic pathologies), the stage of shock (including the extent of end-organ failure), and the patient’s age and beliefs along with the goals of care.

In this context, a specific timeline for “early phenotypization”, “prompt management”, and “exit strategies” should always be implemented in these patients devoid of chances for native heart recovery.

## Figures and Tables

**Figure 1 jpm-12-01463-f001:**
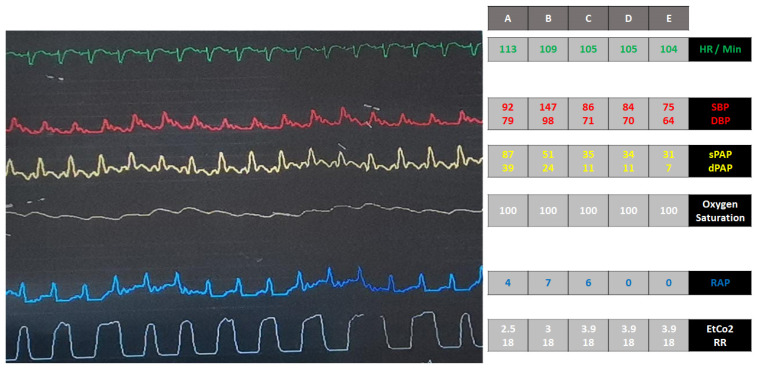
Hemodynamic profiles after Impella placement. Hemodynamics deterioration (panel **A**); normalization of pulmonary pressures and hemodynamics stabilization after the patient started mechanical ventilation with inhalation of 20 parts per million of nitric oxide (panel **B** and **C**); clinical picture deterioration (panel **D** and **E**) being refractory to further increase of inotropes. HR = heart rate; SBP = systolic blood pressure; DBP = diastolic blood pressure; sPAP = systolic pulmonary artery pressure; dPAP = diastolic pulmonary artery pressure; RAP = right atrial pressure; EtCO_2_ = end-tidal CO_2_; RR = respiratory rate.

**Figure 2 jpm-12-01463-f002:**
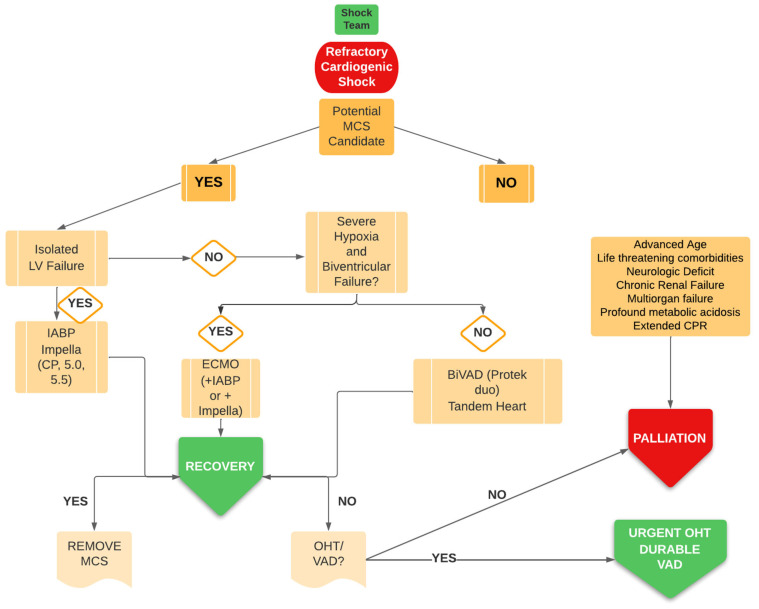
Decisional algoritm in CS. LV = left ventricle; ECMO = venoarterial extracorporeal membrane oxygenation; IABP = intraortic balloon counterpulsation.

**Table 1 jpm-12-01463-t001:** Clinical variables and parameters to define society for cardiovascular angiography and interventions Stages [6]. SBP= systolic blood pressure; MBP= mean blood pressure; ALT = alanine aminotransferase; RAP = right atrial pressure; PAWP = pulmonary arterial wedge pressure; SCAI = Society for Cardiac Angiography and Interventions; SNP = sodium nitroprusside; NA = not applicable.

Variables	Day 1	Day 6	Day 12	Day 13	Day 18	Day 18	Day 19
SBP, (mmHg)	110/60	90/50	110/60	85/40	70/40	100/60	NA
MAP, mmHg	76	63	76	55	50	63	40
Lactate, mmol/L	2.18	2.5	1.8	3	4	2.2	6.4
ALT, U/L	63	68	32	46	55	NA	73
RAP, mmHg	12	16	5	14	16	0	0
PAWP, mmHg	NA	NA	15	NA	18	9	NA
SCAI stages	C	C	C	D	D	D	E
Treatment Intensity	Dopamine; SNP	Epinephrine; IABP	Epinephrine MD; weaning of IABP	Epinephrine; IABP	Epinephrine; IABP	Epinephrine; Impella CP	Epinephrine; Impella CP

## Data Availability

Data available on request due to restrictions (privacy).

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
