# Peer review of "Left Ventricular Unloading in Acute on Chronic Heart Failure: From Statements to Clinical Practice"

_jpm, 2022, doi:10.3390/jpm12091463_

Round 1
Reviewer 1 Report
I would like to congratulate the authors for this highly interesting and relevant “case-review” considering left ventricular unloading in acute on chronic heart failure. The design of the review is innovative and well conducted, with fairly adequate teaching points as presented in final questions.
The manuscript is overall well written with explicative figures.
Addressing the following points will further ameliorate the readability and the value to the manuscript
Minor Comment:
- Minor spelling errors are present along the manuscript, please revise them.
- Page 8, line 262, some reference is missing “ This 260 could have allowed resolution of organ damage and probably early extubation and mo- 261 bilization (25lettera nuccia). “
Author Response
REVIEWER 1
I would like to congratulate the authors for this highly interesting and relevant “case-review” considering left ventricular unloading in acute on chronic heart failure. The design of the review is innovative and well conducted, with fairly adequate teaching points as presented in final questions.
The manuscript is overall well written with explicative figures.
Addressing the following points will further ameliorate the readability and the value to the manuscript
We thank the reviewer for his/her comments and suggestions.
Minor Comment:
- Minor spelling errors are present along the manuscript, please revise them. We have revised the typos and spelling errors
- Page 8, line 262, some reference is missing “This 260 could have allowed resolution of organ damage and probably early extubation and mo- 261 bilization (25lettera nuccia). We did add the reference missing and correct reference 25 (which is now 28 because we added extra references)
Reviewer 2 Report
The aim of the study was to overview the main pit falls which can occur during patients’ management with temporary mechanical circulatory support through a clinical case that illustrates the strict interdependency between left ventricular unloading and right ventricular dysfunction in case of low filling pressures.
The work is well-written; however, some changes are required before acceptance of the manuscript as discussed in the following:
1. Key words should be written as Keywords; moreover, every keyword must start with a capital letter.
2. Figure 1 is not clear as it’s very difficult to read it, make it clear.
3. Abstract needs to be more comprehensive.
4. There are good pieces of information in the case selection but not presented in a well-mannered. Try to present the data in some systematic form.
5. How these questions were designed at the end of the discussion, satisfy with the provision of sufficient data.
6. Discussion should be supported with more relevant references.
7. Any limitations of this case study?
Author Response
Please see the attachment if you have problems with the box below
REVIEWER 2
We thank the reviewer for the comments and suggestions which, we think, have improved the manuscript.
You will find reviewer comments in bold and author's answers in italic.
- Key words should be written as Keywords; moreover, every keyword must start with a capital letter. We did appropriately correct typos following your indication.
- Figure 1 is not clear as it’s very difficult to read it, make it clear. See point 4.
- Abstract needs to be more comprehensive. Abstract was changed to highlight that ADHF-CS and MI-CS are two different scenarios and that there is an unmet need to identify adequate strategies for the former clinical setting. In particular ADHF-CS can be more deceitful at presentation.
MODIFIED ABSTRACT
Abstract
Cardiogenic shock remains a deadly complication of acute on chronic decompensated heart failure (ADHF-CS). Despite its increasing prevalence, it is incompletely understood and therefore often misdiagnosed in the early phase. Precise diagnosis of the underlying cause of CS is fundamental for undertaking the correct therapeutic strategy. Temporary mechanical circulatory support (tMCS) is the mainstay of management: identifying and selecting optimal patients through understanding of the hemodynamics and a prompt profiling and timing, is key for success. A recent statement from the American Heart Association has provided pragmatic suggestions on tMCS device selection, escalation, and weaning strategies. However, several areas of uncertainty still remain in clinical practice. Accordingly, we present an overview of main pitfalls which can occur during patients’ management with tMCS through a clinical case. This case illustrates the strict interdependency between left ventricular unloading and right ventricular dysfunction in case of low filling pressures. Moreover, it further illustrates the pivotal role of stepwise escalation of therapy in a patient with an ADHF-CS and its peculiarities as compared to other forms of acute heart failure.
- There are good pieces of information in the case selection but not presented in a well-mannered. Try to present the data in some systematic form. The case data were better presented with an added table (table 1). We did delete figure 1.
|
Variables |
Day 1 |
Day 6 |
Day 12 |
Day 13 |
Day 18 |
Day 18 |
Day 19 |
|
SBP, (mmHg) |
110/ 60 |
90/50 |
110/60 |
85/40 |
70/40 |
100/60 |
NA |
|
MAP, mmHg |
76 |
63 |
76 |
55 |
50 |
63 |
40 |
|
Lactate, mmol/L |
2.18 |
2.5 |
1.8 |
3 |
4 |
2.2 |
6.4 |
|
ALT, U/L |
63 |
68 |
32 |
46 |
55 |
NA |
73 |
|
RAP, mmHg |
12 |
16 |
5 |
14 |
16 |
0 |
0 |
|
PAWP, mmHg |
NA |
NA |
15 |
NA |
18 |
9 |
NA |
|
SCAI stages |
C |
C |
C |
D |
D |
D |
E |
|
Treatment Intensity |
Dopamine; SNP |
Epinephrine; IABP |
Epinephrine, Weaning of IABP |
Epinephrine; IABP |
Epinephrine; IABP |
Epinephrine; Impella CP |
Epinephrine; Impella CP |
- How these questions were designed at the end of the discussion, satisfy with the provision of sufficient data. We have chosen this format to write a review with clear and practical teaching points (as also suggested by reviewer 1).
- Discussion should be supported with more relevant references. We thank the reviewer for this suggestion: we added recent works to further support our introduction, discussion and teaching points (see reference number 6, 13, 14, 32 and 33)
- Any limitations of this case study? As we have highlighted at teaching point 3, we think that the lack of invasive continuous monitoring was the main limitation at the moment of hemodynamic deterioration. As a result, we did overlook the fact that IABP (second placement) was not enough for unloading and uneffectivel for circulatory support.

Reviewer 3 Report
The authors of the current manuscript present an overview of the main pitfalls, which can occur during management of patients in cardiogenic shock, in which temporary mechanical circulatory support had been applied.
The case is interesting for the clinicians, however, it is not a case of a rare disease/entity with an unclear (unsolved) diagnostic/therapeutic approach. A single case is not a base for general recommendation/guidelines - as the authors say in the beginning, cardiogenic shock is with "increasing incidence". I recommend to the authors to collect more cases like this one and to present an analysis of case series or a clinical study which will validate scientifically their work.
Author Response
We thank the reviewer for the comments; however, we did choose one clinical case because it offers many neglected issues which might improve outcomes. We did not aim at collecting cases of standard management and, fortunately, we did not have any other case like the one presented. We are aware that this case does not describe a rare condition, but we have just chosen it for this reason: though very frequent in our ICCUs, ADHF-CS is considerably less understood compared to MI-CS, both as far as it concerns identification and outcomes. Probably one of the most important and often underappreciated means to learn from our complications is to publish them. It is with dissemination and discussion of these mistakes and how they were managed that we may learn from each other.
Round 2
Reviewer 3 Report
The authors of the current manuscript have improved their scientific paper, observing most of the reviewers' recommendations.
The manuscript can be published in its current version.